**Assessing snow extent data sets over North America to inform and improve trace gas**
**retrievals from solar backscatter**
Matthew J. Cooper[1], Randall V. Martin[1,2], Alexei I. Lyapustin[3], and Chris A. McLinden[4]
1. Department of Physics and Atmospheric Science, Dalhousie University, Halifax, Nova Scotia,
Canada.
2. Harvard-Smithsonian Center for Astrophysics, Cambridge, Massachusetts, USA
3. NASA Goddard Space Flight Center, Greenbelt, MD, USA
4. Air Quality Research Division, Environment and Climate Change Canada, Toronto, Ontario,
Canada
**Abstract**
Accurate representation of surface reflectivity is essential to tropospheric trace gas retrievals
from solar backscatter observations. Surface snow cover presents a significant challenge due to
its variability and thus snow-covered scenes are often omitted from retrieval data sets; however,
the high reflectance of snow is potentially advantageous for trace gas retrievals. We first
examine the implications of surface snow on retrievals from the upcoming TEMPO
geostationary instrument for North America. We use a radiative transfer model to examine how
an increase in surface reflectivity due to snow cover changes the sensitivity of satellite retrievals
to $NO_2$ in the lower troposphere. We find that a substantial fraction (>50%) of the TEMPO field
of regard can be snow covered in January, and that the average sensitivity to the tropospheric
$NO_2$ column substantially increases (doubles) when the surface is snow covered.
We then evaluate seven existing satellite-derived or reanalysis snow extent products against
ground station observations over North America to assess their capability of informing surface
conditions for TEMPO retrievals. The Interactive Multisensor Snow and Ice Mapping System
(IMS) had the best agreement with ground observations (accuracy=93%, precision=87%,
recall=83%). Multiangle Implementation of Atmospheric Correction (MAIAC) retrievals of
MODIS observed radiances had high precision (90% for Aqua and Terra), but underestimated
the presence of snow (recall=74% for Aqua, 75% for Terra). MAIAC generally outperforms the
standard MODIS products (precision=51%, recall=43% for Aqua; precision=69%, recall=45%
for Terra). The Near-real-time Ice and Snow Extent (NISE) product had good precision (83%)
but missed a significant number of snow covered pixels (recall=45%). The Canadian
Meteorological Centre (CMC) Daily Snow Depth Analysis Data set had strong performance
metrics (accuracy=91%, precision=79%, recall=82%). We use the $F$ score, which balances
precision and recall, to determine overall product performance ($F$ = 85%, 82(82)%, 81%, 58%,
46(54)% for IMS, MAIAC Aqua(Terra), CMC, NISE, MODIS Aqua(Terra) respectively) for
providing snow cover information for TEMPO retrievals from solar backscatter observations.
We find that using IMS to identify snow cover and enable inclusion of snow-covered scenes in
clear-sky conditions across North America in January can increase both the number of
observations by a factor of 2.1 and the average sensitivity to the tropospheric $NO_2$ column by a
factor of 2.7.

## 1.  Introduction

Satellite observations of solar backscatter are widely used as a source of information on

atmospheric trace gases (Richter and Wagner, 2011). These observations have provided valuable
information on vertical column densities of $O_3$, $NO_2$, $SO_2$, CO, HCHO, $CH_4$ and other important
trace gases in the troposphere (Fishman et al., 2008). Satellite observations of trace gases have
been used to assess air quality (Duncan et al., 2014; Martin, 2008) and to gain insight into
atmospheric processes including emissions (Streets et al., 2013), lifetimes (Beirle et al., 2011;
Fioletov et al., 2015; de Foy et al., 2015; Valin et al., 2013), and deposition (Geddes and Martin,
2017; Nowlan et al., 2014). The utility of these observations is dependent on their quality, and
thus ensuring retrieval accuracy is essential.

Previous studies have found that retrieved $NO_2$ vertical column densities are highly

sensitive to errors in assumed surface reflectance (Boersma et al., 2004; Lamsal et al., 2017;
Martin et al., 2002). Much of this error sensitivity results from observation sensitivity to trace
gases in the lower troposphere. The observation sensitivity is accounted for in the air mass factor
(AMF) conversion of observed line-of-sight "slant columns" to vertical column densities.
Uncertainties in surface reflectance are a significant contributor to AMF uncertainty.

Existing reflectivity climatologies (e.g. Kleipool et al., 2008; Koelemeijer et al., 2003; Liang et al., 2002; Herman and Celarier, 1997) do not represent snow cover well, since the statistical methods to exclude reflective clouds from the climatologies also exclude variable snow cover; Correspondingly, surface snow may be mistaken for cloud, leading to errors in cloud fraction and pressure estimates used in trace gas retrievals (Lin et al., 2015; O'Byrne et al., 2010; Vasilkov et al., 2017). Therefore, snow cover is particularly challenging to retrievals. Misrepresenting surface snow cover can lead to large errors (20-50%) in retrieved $NO_2$ columns over broad regions with seasonal snow cover (O'Byrne et al., 2010). For this reason, observations over snow are often omitted or flagged as unreliable to avoid potential errors. This limits the ability of satellite retrieved data sets to offer adequate temporal and spatial sampling in winter months. Additionally, over highly reflective surfaces such as snow observation sensitivity to the lower troposphere is larger and has less dependence on *a priori* $NO_2$ profiles (Lorente et al., 2017; O'Byrne et al., 2010); Thus, omitting snow-covered scenes means omitting the observations with the greatest sensitivity to the lower troposphere. This could be remedied by using a product that would allow for snow cover identification to be done with confidence.

Several data products provide information on snow extent using surface station observations, satellite observed radiances, or visible imagery. Previous evaluations have found it difficult to determine which of these products is definitively the best, partly due to differences in resolution. Most products are more consistent during the winter months when persistent, deep snow is present (Frei et al., 2012; Frei and Lee, 2010). However, disagreements are common during accumulation and melting seasons, over mountains, and under forest canopies. These evaluations have largely focused on local or regional snow cover, or included only cloud-free observations.

The upcoming geostationary Tropospheric Emissions: Monitoring of Pollution (TEMPO) satellite instrument will provide hourly observations of air quality relevant trace gases over North America at an unprecedented spatial and temporal resolution (Zoogman et al., 2017). As is the case for all nadir satellite retrievals, the quality of these observations will depend on the accuracy of the surface reflectance used in the retrieval. As a significant portion of the observed domain experiences snow cover, an accurate representation of snow cover is needed. Current plans to deal with snow cover for TEMPO are to rely on external observations.

87  In this work, we examine the importance of accurate snow identification by using a

88 radiative transport model to evaluate how the vertical sensitivity of a satellite retrieval is

89 impacted by surface reflectance. We then assess seven snow extent products that are expected to

90 continue to be operational during the TEMPO mission using in situ observations across North

91 America with the intent of determining which product is best suited for providing snow cover

92 information for TEMPO and other future satellite retrievals. Finally, we combine radiative

93 transfer model results with a snow extent product to show how including snow-covered scenes

94 improves both the quantity and quality of information in a retrieval data set.

## 96 2. Data and Algorithms

### 97  2.1. Gridded Snow Products

### 98  2.1.1. IMS

99  One of the most widely used sources of snow extent data is the Interactive Multisensor

100 Snow and Ice Mapping System (IMS). IMS provides daily, near-real-time maps of snow and sea

101 ice cover in the Northern Hemisphere at 4km resolution (Helfrich et al., 2007). The maps are

102 produced by a trained analyst using visible imagery from a collection of geostationary (e.g.

103 GOES, MeteoSat) and polar orbiting (e.g. AVHRR, MODIS, SAR) satellite instruments, with

104 additional information from microwave sensors (e.g. DMSP, AMSR, AMSU), surface

105 observations (e.g. SNOTEL), and models (e.g. SNODAS) (Helfrich et al., 2007). By using

106 multiple sources of information with different spatial resolution and temporal sampling, IMS can

107 minimize interference from clouds.

### 108  2.1.2. MODIS

109  A second commonly used snow and ice product is derived from MODIS satellite

110 observations from the Terra and Aqua satellites (Hall and Riggs, 2007). Terra and Aqua have

111 sun-synchronous, near polar orbits with overpass times of 1030 and 1330 hr respectively. Snow

112 cover is calculated using a Normalized Difference Snow Index (NDSI), which examines the

113 difference between observed radiation at visible wavelengths (where snow is highly reflective)

114 and short IR wavelengths (where there is little reflection from snow). Observations are made at

500 m spatial resolution and aggregated to produce daily snow cover fractions on a 0.05°
resolution grid. Past evaluations of the standard MODIS snow product show good agreement in
cloud-free conditions but often snow is misidentified as cloud (Hall and Riggs, 2007; Yang et al.,

2015).

The Multiangle Implementation of Atmospheric Correction (MAIAC) algorithm is

another algorithm processing MODIS observations. MAIAC retrievals uses radiances observed
by the MODIS Aqua and Terra satellites to provide atmospheric and surface products including
snow detection on a 1 km grid (Lyapustin et al., 2011a, 2011b, 2012). While the NDSI used by
the standard MODIS product is also used by MAIAC as one of the criteria, the overall snow and
cloud detection in MAIAC are different from the standard MODIS algorithm (Lyapustin et al.,

2008).

**2.1.3. NISE**

The Near-real-time Ice and Snow Extent (NISE) provides daily updated snow cover

extent information on a 25x25 km grid (Nolin et al., 2005). NISE uses microwave measurements
from the Special Sensor Microwave Imager/Sounder (SSM/I) on a sun-synchronous, quasi-polar
orbit to observe how microwave radiation emitted by soil is scattered by snow. Products based
on microwave measurements such as NISE are known to miss wet and thin snow, as wet snow
emits microwave radiation similar to soil, and thin snow does not provide sufficient scattering.

**2.1.4. CMC**

The Canadian Meteorological Centre (CMC) Daily Snow Depth Analysis Data is a

statistical interpolation of snow depth measurements from 8,000 surface sites across Canada and
U.S. interpolated using a snow pack model (Brasnett, 1999). Unlike the aforementioned satellite
products that provide snow extent, CMC provides snow depths. Daily snow maps are produced
at 25 km resolution. As it a reanalysis product, there is a time delay in availability. The CMC
snow depths show good agreement with independent observations over midlatitudes and is
considered an improvement over previous snow depth climatologies (Brown et al., 2003).

**2.2 Surface observations**

These snow identification products are evaluated against surface station observations
from the Global Historical Climatology Network-Daily (GHCN-D) database, an amalgamation
of daily climate records from over 80,000 surface stations worldwide (Menne et al., 2012a).
Most observations over Canada and the United States are collected by government organizations
(Environment and Climate Change Canada and NOAA National Climatic Data Center,
respectively) with additional measurements from smaller observation networks. While the focus
of the database is collecting temperature and precipitation measurements, many stations (1,279 in
Canada, 13,932 in United States in 2015 used here) also offer snow depth measurements.
A subset of the surface stations included in GHCN-D may also be used in the CMC
reanalysis. It is difficult to definitively know which stations are used, as CMC does not routinely
archive this information. However, we estimate that only 5% of the GHCN-D stations used here
are located within $0.1°$ of a possible CMC station, and thus GHCN-D has sufficient independent
information sources to evaluate the CMC product.
**2.3 Radiative transfer calculations**
The sensitivity of satellite observations of $NO_2$ to its vertical distribution is calculated
here using the LIDORT radiative transfer model (Spurr, 2002). The model is used to calculate
scattering weights, which quantify the sensitivity of backscattered solar radiation to $NO_2$ at
different altitudes (Martin et al., 2002; Palmer et al., 2001). The observation sensitivity to lower
tropospheric $NO_2$ is represented by the air mass factor. Air mass factors for OMI satellite
observations in January 2013 are calculated as a useful analog for future TEMPO observations as
both instruments are spectrometers observing reflected sunlight at UV to visible wavelengths.
AMFs are calculated at 440 nm, at the centre of the $NO_2$ retrieval window for OMI and TEMPO
where $NO_2$ has strong absorption features. Vertical $NO_2$ profiles, and other trace gas and aerosol
profiles needed for the AMF calculation shown here, are obtained from a simulation of the
GEOS-Chem chemical transport model version 11-01 (www.geos-chem.org).
Figure 1 shows maps of snow-free and snow-covered reflectances used here. Snow-free
surface reflectance at 470 nm is provided by Nadir BRDF-Adjusted reflectances from the
MODIS CMG Gap-Filled Snow-Free Products (Sun et al., 2017). Reflectivities at 354 nm for
snow-covered scenes are derived from OMI observations as described by O'Byrne et al. (2010).
This data set is consistent with previous snow reflectivity (e.g. Moody et al., 2007; Tanskanen
and Manninen, 2007) over most land types (O'Byrne et al., 2010). Snow-covered reflectivity has
an estimated uncertainty of 10-20% in most regions, with higher uncertainties in regions with
thin or transient snow. Although the 354 nm wavelength is different than the 440 nm wavelength
used to calculate AMFs, snow reflectivity has weak spectral dependence in UV-Visible
wavelengths (Feister and Grewe, 1995; O'Byrne et al., 2010). Snow can increase surface
reflectance by over a factor of 10 in central North America where short vegetation is readily
covered by snow.

## 3. Methods

Here we test daily snow cover products for 2015. Snow products are regridded from their
native resolutions to a common 4 km grid (similar to the spatial resolution of TEMPO). A grid
box is considered to be snow covered if any observations within that box are snow covered.
MAIAC, NISE, and IMS give only a yes/no flag for presence of snow. MODIS products provide
a pixel snow fraction, and we consider any pixels with nonzero snow fractions as snow covered.
Any CMC grid box with nonzero snow depth is considered snow covered.
GHCN-D surface measurements are used as the ground "truth" for evaluating the satellite
and reanalysis snow data products tested here. If measurements from multiple surface data
networks exist in the same grid box, the most reliable source is used per the priority order given
by GHCN-D (Menne et al., 2012b). If observations from multiple surface stations within the
most reliable network within a grid box disagree on the presence of snow on a given day, that
day is excluded from the evaluation.
We assess the snow data sets using metrics that are commonly used for evaluating binary
data sets (Rittger et al., 2013). These metrics are based on the possible outcomes for identifying
snow: true positive (TP), true negative (TN), false positive (FP), and false negative (FN).
Accuracy measures the likelihood that a grid box, with snow or without, is correctly classified:

$$Accuracy = \frac{TP + TN}{TP + TN + FP + FN} \tag{1}$$

Precision is the probability that a region identified as snow-covered has snow:

$$Precision = \frac{TP}{TP + FP} \tag{2}$$

Recall is the likelihood that snow cover is detected when present:

$$Recall = \frac{TP}{TP + FN} \tag{3}$$

The *F* score balances recall (which accounts for false negatives) and precision (which accounts
for false positives) to measure correct classification of snow without the influence of frequent
snow-free periods, and therefore is the metric which is most relevant for TEMPO:

$$F = 2 * \frac{precision * recall}{precision + recall} \tag{4}$$

**4. Results**
We first examine the effect of surface reflectivity on retrieval sensitivity by using the
LIDORT radiative transfer model to calculate $NO_2$ air mass factors for both snow-free and snow-
covered scenarios using the corresponding snow-free (Sun et al., 2017) or snow-covered
(O'Byrne et al., 2010) surface reflectance over North America. We calculate air mass factors
over North America in January 2013. We assume cloud-free conditions in all AMF calculations,
as the impact of surface reflectance on retrieved cloud fractions is beyond the scope of this
paper.
Figure 2 shows the sensitivity of backscattered radiation (scattering weights) over snow-
covered and snow-free surfaces for two locations; a midlatitude location (US Midwest, 42°N,
99°W) with a solar zenith angle of 60° and at a high latitude location (Northern Canada, 58°N,
76°W) with a solar zenith angle of 79°. The snow-covered scattering weights are greater than the
snow-free scattering weights throughout the troposphere, by factors of 2.0 (2.7) below 5 km, 2.7
(3.7) below 2 km, and 2.6 (5.3) below 1 km at the mid (high) latitude location. This shows that
satellite observed backscattered radiation in clear-sky conditions is up to five times as sensitive
to $NO_2$ in the boundary layer after accounting for increased reflection by snow, due to the
increased absorption by $NO_2$ in the lower troposphere when the surface reflects more sunlight.
Figure 3 shows the distribution of AMF values over North America with and without
reflectance from snow. The snow-free AMF distribution is unimodal with a median of 1.2.
Allowing for the presence of snow introduces a second mode with a median of 3.2. Mean AMFs
increase by a factor 2.0 in the presence of snow, indicating an overall doubling in the sensitivity
to tropospheric $NO_2$ over snow covered surfaces across North America. The impact is larger over
polluted regions, as mean AMFs increase by a factor of 2.2 in regions where $NO_2$ columns
exceed $1x10^{15}$ molec/cm$^2$. Maps of AMF with and without snow cover for January 2013 show
that AMF values increase over 69% of the land surface within the TEMPO domain.

We next examine the snow datasets to identify the one most suited for the TEMPO
retrieval algorithm. Figure 4 shows the spatial distribution of false positives and false negatives
in the data sets. In all data sets, both false positives and negatives are most frequent over
mountainous regions, particularly in the Rocky Mountain region, consistent with previous
validation studies (Chen et al., 2012, 2014; Frei et al., 2012; Frei and Lee, 2010). These errors
are often attributed to differences in representativeness, as snow cover in mountain regions is
often spatially inhomogeneous, and thus *in situ* measurements may not be representative of the
pixel. A slight increase in the number of false positives in IMS over mid-western and prairie
regions may result from crop regions with high snow-free albedos being mistaken for snow in
visible imagery (Chen et al., 2012; Yang et al., 2015). NISE, MODIS Aqua, and MODIS Terra
have more false negatives overall, especially in the Great Lakes and New England regions. False
positives are less frequent than false negatives in all data sets. IMS and CMC have the lowest
frequency of false negatives. NISE and MAIAC have the lowest frequency of false positives.

Figure 5 shows the metrics used to evaluate data set performance. Table 1 summarizes
these results. All data sets have high accuracy numbers, owing largely to a high number of true
negatives during the summer months. MODIS Aqua and Terra have low recall and *F* scores.
When only observations with MODIS cloud fractions less than 20% are used, MODIS has better
agreement with the ground stations (*F* statistic increases from 0.38 to 0.49 at native resolution
for Aqua, 0.43 to 0.63 for Terra), however this reduces the number of usable MODIS
observations by up to 60%. NISE has high precision but low recall, indicating that while areas
classified as snow-covered by NISE are likely correct, many snow-covered regions are missing
in the data set. This is consistent with evaluations by McLinden et al. (2014) and O'Byrne et al.
(2010). Although CMC, IMS, and MAIAC products show an increase in frequency of false
negatives over the Rocky Mountains, they retain a high precision in this region due to frequent
snow cover. While MAIAC Aqua/Terra have high accuracy and precision, lower recall values
indicate that they are conservative in identifying the presence of snow. This is possibly a
consequence of the method used for identifying cloud, which may incorrectly classify fresh
snowfall as cloud (Lyapustin et al., 2008). Data sets were also evaluated by season with similar
results (Appendix Table A1). All data sets have weaker performance metrics during the spring
melt season, which has been observed in past evaluations (Frei et al., 2012). IMS has the highest
*F* score in winter and autumn but is slightly outperformed by MAIAC in spring. Data sets were
also evaluated at their native resolutions and at a common 25 km resolution (Appendix Tables
A2-3). Results are similar at each resolution with two exceptions: MODIS Aqua and Terra
products perform better when regridded from their native 0.05° resolution to a 4 km resolution as
it reduces the number of grid boxes missing observations due to cloud, and MAIAC Aqua and
Terra perform better at their native resolution than at either 4 km or 25 km as degrading the
spatial resolution results in a loss of information.

For all data sets, recall is generally low in two regions: along the Pacific coastline where
snow depths are relatively thin, and in the south when snow is rare and generally short lived.
Thin snow is likely to be less homogenous across a pixel and more likely to be obscured by
forest canopies or tall grasses, and thus is difficult to observe from satellite imagery. Short lived
snow in the south is likely to be missed by satellite observations, especially since clouds are
often present. However, as IMS uses multiple observations at multiple times of day in addition to
incorporating ground station data, it is more likely to find snow in these cases than other satellite
products (Hall et al., 2010). Overall, IMS has best agreement with *in situ* observations, with the
highest accuracy, recall, and *F* statistic and relatively high precision.

While CMC also has strong performance metrics, it is important to consider the
information source used to describe snow extent in each product. Products based on satellite
observations are advantageous when assessing how surface reflectivity affects backscattered
radiation observed from space. For example, thin snow, or snow obscured by tree canopies, may
not affect the observed brightness from space, but would be considered snow-covered by a
product based on surface observations (e.g. CMC). Also, the reflectivity of a snow-covered
surface decreases over time as the snow ages (Warren and Wiscombe, 1980); This effect would
not be captured by snow depth measurements. And while snow depth has been used as an
indicator of brightness (Arola et al., 2003), it can not account for snow aging or canopy effects.
IMS is based on visible satellite imagery and thus determines snow extent based on brightness
from space, which is more applicable to satellite retrievals. And while most satellite-based
products rely on observations made at a single overpass time and viewing geometry, IMS has the
advantage of incorporating observations from multiple satellites with differing measurement
times and geometries, including both geostationary and low Earth orbits. These reasons, in
addition to a strong agreement with in situ measurements and near-real-time updates, make IMS
best suited for informing TEMPO retrievals.
We next examine the effect on both spatial sampling and sensitivity to the lower
troposphere of a retrieval data set if observations with surface snow are included rather than
omitted. We use IMS to identify the presence of snow for OMI observations over North America
in January 2015. We then use LIDORT to calculate AMFs for these observations using the
corresponding snow-free (Sun et al., 2017) or snow-covered (O'Byrne et al., 2010) surface
reflectance, and examine the results of either including or omitting snow-covered scenes. Figure
6 shows that including snow-covered scenes results in a significant (factor of 2.1) increase in
observation frequency, particularly in the northern US and Canada. Additionally, including
snow-covered scenes increases the average AMF by a factor of 2.7 in regions with occasional
snow cover. The increase in AMF demonstrates that including snow-covered scenes increases
the quality of information about the tropospheric $NO_2$ column by increasing the observation
sensitivity to tropospheric $NO_2$. As we assume clear-sky conditions, these are likely upper
bounds on potential increases in observation quantity and quality. In practice, the presence of
clouds and errors in cloud retrieval algorithms will likely diminish these impacts.

**5. Conclusion**

An accurate representation of snow cover is essential to ensuring satellite retrieval
accuracy, including those from TEMPO. Radiative transfer model calculations indicate that
clear-sky $NO_2$ retrievals over reflective snow-covered surfaces are more than twice as sensitive
to $NO_2$ in the boundary layer than over snow-free surfaces. This makes snow an attractive
surface over which to observe tropospheric $NO_2$. However, the lack of confidence in snow
identification has previously led many retrieval procedures to omit observations over snow. We
show that increasing this confidence such that these observations could be included not only
improves spatial and temporal sampling, but also allows the inclusion of observations with
higher quality information on the lower troposphere.

We evaluated seven snow extent data sets to determine their usefulness for informing satellite retrievals of trace gas from solar backscatter observations. All products were more likely to misidentify snow over mountains or where snow cover is thin or short lived. IMS had the best agreement with *in situ* observations ($F=0.85$), and as a satellite based, operational, daily updated product, it is well suited for informing TEMPO satellite retrievals. The low recall value (0.45) for NISE indicated that a significant number of snow covered pixels are missed. The standard MODIS products showed medium precision and low recall owing to cloud contamination. The MAIAC products had the highest precision (0.90 for both Aqua and Terra) of those tested, but is conservative in ascribing the presence of snow (recall=0.74 for Aqua, 0.75 for Terra). CMC had strong performance metrics ($F=0.81$), but as a reanalysis product based on ground observations it may not appropriately represent how a surface snow reflectivity would affect TEMPO observed radiances.

The potential improvements in $NO_2$ retrieval performance over snow-covered scenes outlined here were tested for clear-sky conditions. The accuracy of cloud retrieval schemes also impacts the quality of trace gas retrievals. Many cloud retrieval schemes have difficulty distinguishing between a bright surface and bright, low altitude clouds; This may diminish the impact that improved surface snow reflectance can have on observation frequency and sensitivity when clouds are present. However, using accurate surface snow cover information may also lead to corresponding improvements in cloud retrieval accuracy.

Future work should investigate snow reflectance products that could be used when snow is detected. This could potentially include Bidirectional Reflectance Distribution Functions (BRDF) that describe reflection at different viewing angles, as this effect has been shown to have significant impact on retrieved $NO_2$ columns and clouds (Lorente et al., 2018; Vasilkov et al., 2017). Accurate knowledge of snow reflectivity is also needed to improve retrievals over snow. A retrieval algorithm that combines daily snow detection from IMS with a climatology of snow reflectance has the potential to greatly improve upon current methodologies.

**6. Data Availability**

IMS (National Ice Center, 2008), NISE (Brodzik and Stewart, 2016), MODIS Aqua (Hall

and Riggs, 2016a), MODIS Terra (Hall and Riggs, 2016b), and CMC (Brown and Brasnett,
2010) data are available from the NASA National Snow and Ice Data Center (http://nsidc.org).
MAIAC Collection 6 re-processing of MODIS data started in September 2017 and is expected to
be completed by the end of year. This study used MAIAC data currently available via ftp at
NASA Center for Climate Simulations (NCCS):
ftp://maiac@dataportal.nccs.nasa.gov/DataRelease/. GHCN-D data are available from the
NOAA National Climatic Data Center (Menne et al., 2012b; www.ncdn.noaa.gov). Code for
calculating scattering weights and air mass factors, and snow-covered surface reflectances used
here, are available at http://fizz.phys.dal.ca/~atmos. Snow-free surface reflectances are available
at ftp://rsftp.eeos.umb.edu/data02/Gapfilled/. The GEOS-Chem chemical transport model used
here is available at www.geos-chem.org.

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

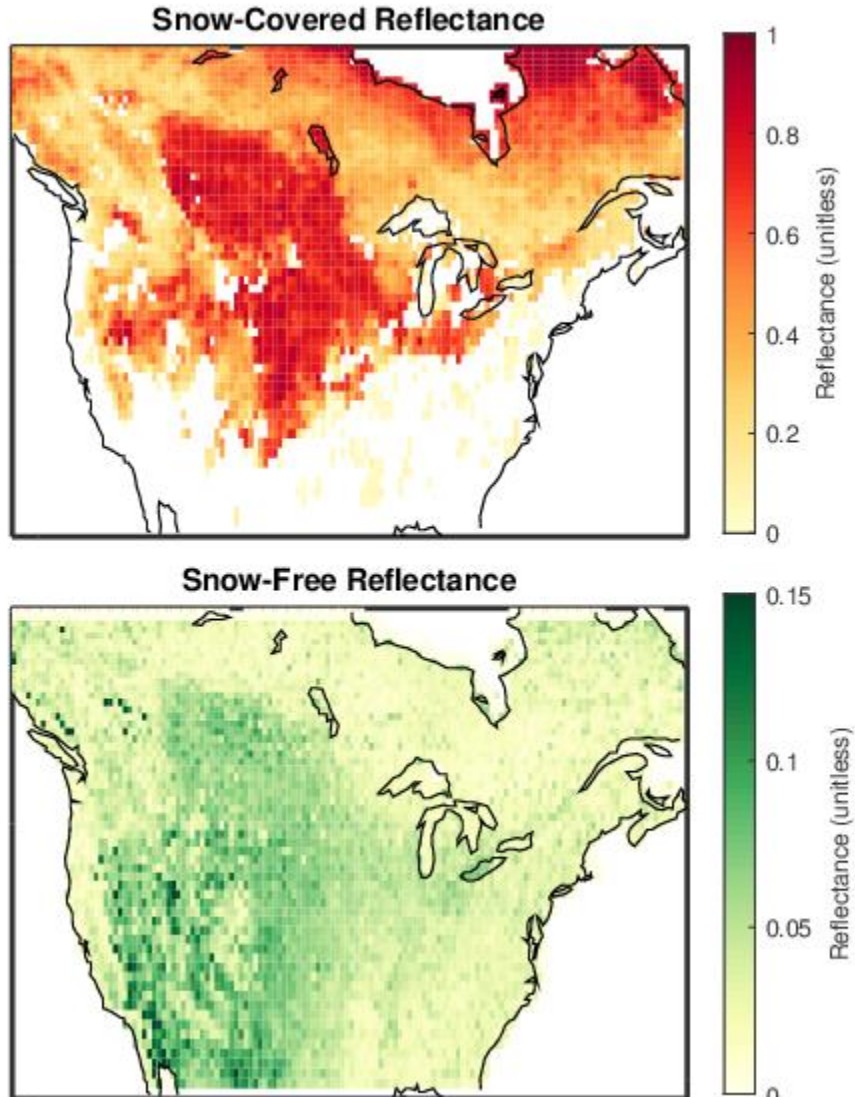


Figure 1: Surface reflectivity at UV-visible wavelengths for snow-covered and snow-free
conditions for January 2013. White space in top panel indicates no snow reflectance information
is available.

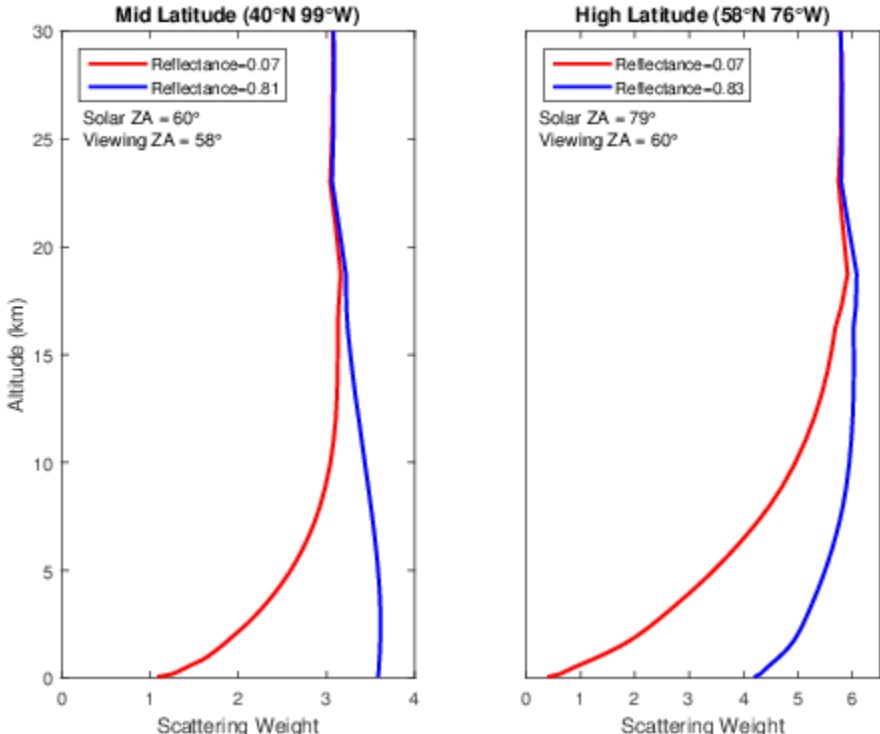


Figure 2: Observation sensitivity to $NO_2$. Scattering weight profiles calculated for cloud-free

OMI $NO_2$ retrievals, with and without surface snow cover, for January 2013 at (Left) 42° N, 99°

W with a solar zenith angle (ZA) of 60° and (Right) 58° N, 76° W with a solar zenith angle of

79°.


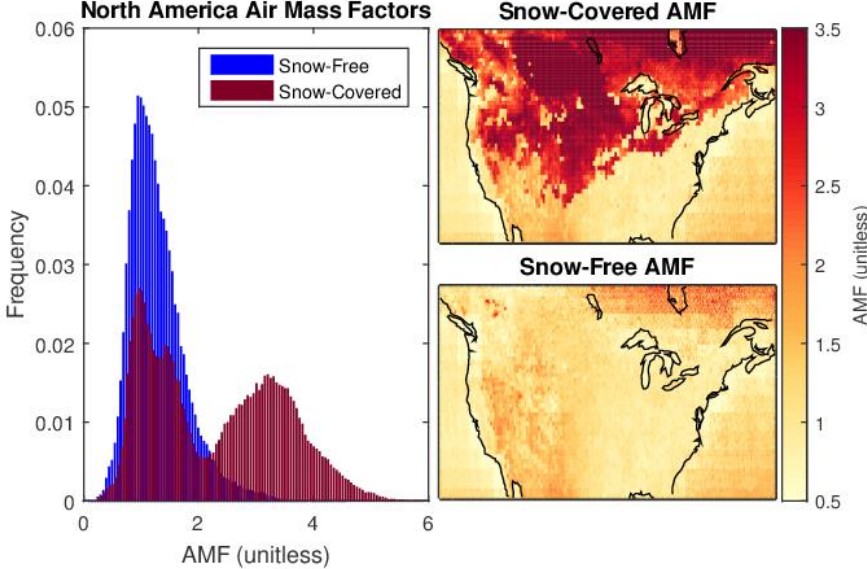


Figure 3: (Left) Distribution of Air Mass Factors (AMFs) calculated for OMI $NO_2$ retrievals over
North America for observation geometry of January 2013, using snow-free (Sun et al., 2017) or
snow-covered (O'Byrne et al., 2010) surface reflectance. (Right) Maps of AMF for snow-
covered and snow-free conditions.

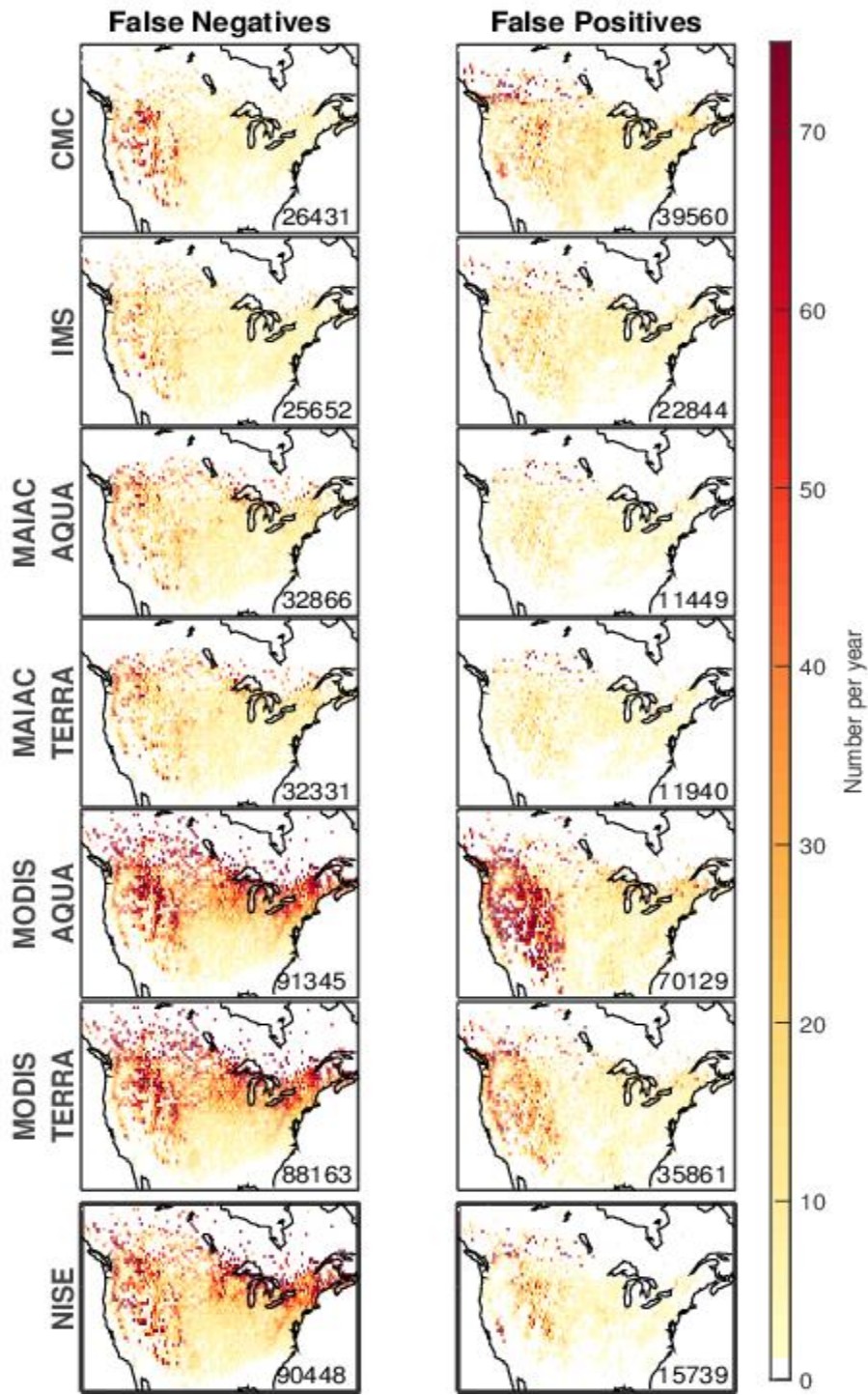


Figure 4: Number of false positive (FP) and false negative (FN) snow attributions by the snow

data sets in 2015. All data sets are evaluated at 4 km resolution. Total number of false snow

attributions inset. White space indicates no ground stations present.

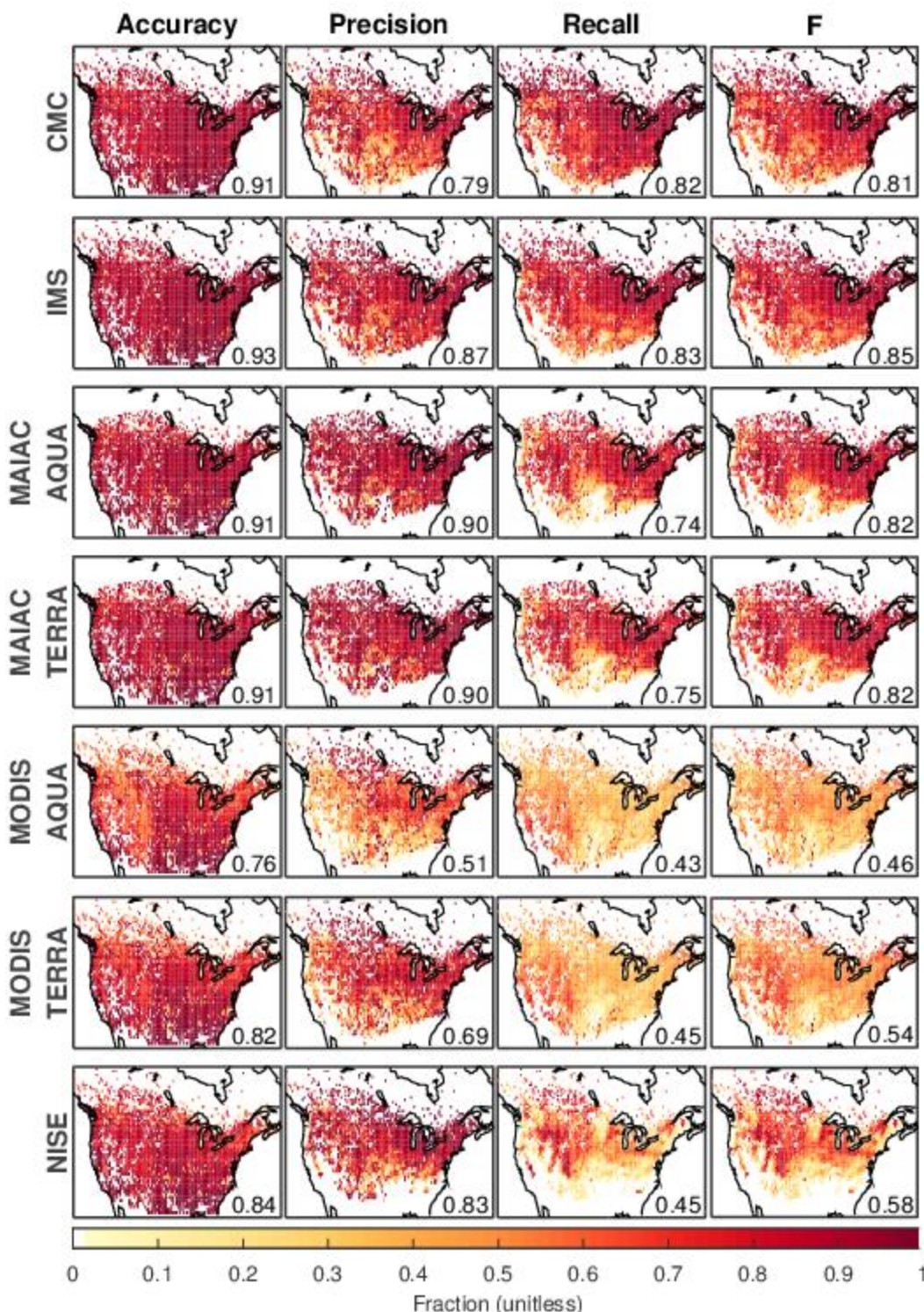


Figure 5: Statistical metrics to evaluate snow cover products. All data sets are gridded at 4 km

resolution. White space indicates no ground stations present.

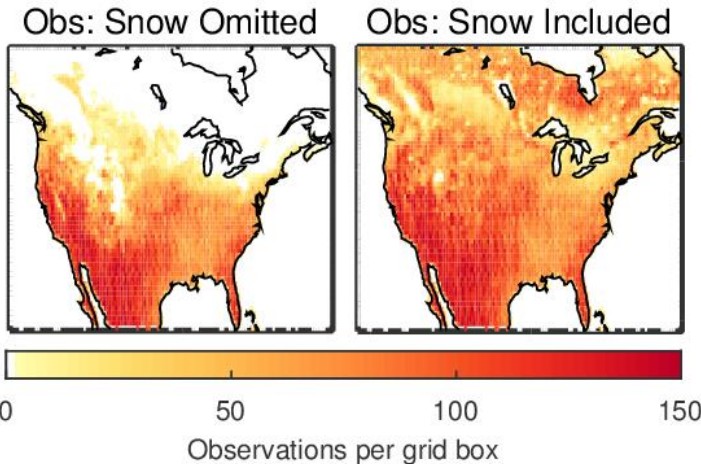

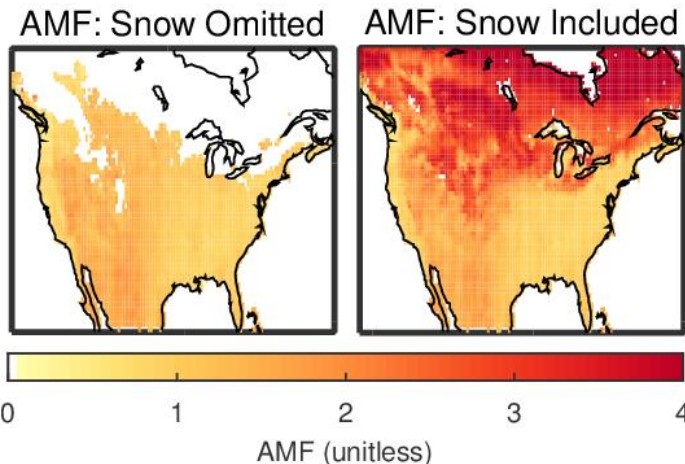


Figure 6: OMI observation frequency (top) and average AMFs (bottom) over North America in
January using IMS to identify surface snow conditions. White space indicates a lack of
observations.


| | Accuracy | Precision | Recall | F |
|---|---|---|---|---|
| CMC | 0.91 | 0.79 | **0.83** | 0.81 |
| IMS | **0.93** | 0.87 | **0.83** | **0.85** |
| MAIAC AQUA | 0.91 | **0.90** | 0.74 | 0.82 |
| MAIAC TERRA | 0.91 | **0.90** | 0.75 | 0.82 |
| MODIS AQUA | 0.76 | 0.51 | 0.43 | 0.46 |
| MODIS TERRA | 0.82 | 0.69 | 0.45 | 0.54 |
| NISE | 0.84 | 0.83 | 0.45 | 0.58 |

Table 1: Evaluation of daily snow extent data set performance for 2015. GHCN-D surface
observations are used as "truth". All products are regridded to a common 4 km resolution. The
highest value for each metric is shown in bold.
**Appendix**

| Months | Data Set | Accuracy | Precision | Recall | F |
|---|---|---|---|---|---|
| | CMC | 0.84 | 0.84 | 0.89 | 0.86 |
| | IMS | **0.88** | 0.90 | **0.88** | **0.89** |
| | MAIAC AQUA | 0.84 | **0.93** | 0.80 | 0.86 |
| DJF | MAIAC TERRA | 0.84 | 0.92 | 0.80 | 0.86 |
| | MODIS AQUA | 0.58 | 0.84 | 0.34 | 0.48 |
| | MODIS TERRA | 0.60 | 0.88 | 0.37 | 0.52 |
| | NISE | 0.63 | 0.90 | 0.41 | 0.57 |
| | CMC | 0.90 | 0.63 | 0.57 | 0.59 |
| | IMS | **0.93** | 0.74 | **0.67** | 0.70 |
| | MAIAC AQUA | **0.93** | **0.81** | 0.62 | **0.71** |
| MAM | MAIAC TERRA | **0.93** | **0.81** | 0.63 | **0.71** |
| | MODIS AQUA | 0.86 | 0.43 | 0.39 | 0.41 |
| | MODIS TERRA | 0.89 | 0.62 | 0.40 | 0.49 |
| | NISE | 0.90 | 0.71 | 0.34 | 0.46 |
| | CMC | 0.91 | 0.73 | **0.81** | 0.76 |
| | IMS | **0.92** | 0.82 | 0.74 | **0.78** |
| | MAIAC AQUA | 0.91 | **0.86** | 0.60 | 0.71 |
| SON | MAIAC TERRA | 0.90 | 0.85 | 0.61 | 0.71 |
| | MODIS AQUA | 0.82 | 0.51 | 0.36 | 0.42 |
| | MODIS TERRA | 0.86 | 0.71 | 0.39 | 0.51 |
| | NISE | 0.85 | 0.85 | 0.25 | 0.39 |

Table A1: Evaluation of daily snow extent data set performance by season for 2015. GHCN-D
surface observations are used as "truth". All products are regridded to a common 4 km
resolution. The highest value for each metric/season is shown in bold.


|  | Resolution | Accuracy | Precision | Recall | F |
|---|---|---|---|---|---|
| CMC | 25 km | 0.92 | 0.81 | 0.81 | 0.81 |
| IMS | 4 km | **0.93** | 0.87 | **0.83** | **0.85** |
| MAIAC AQUA | 1 km | 0.91 | **0.91** | 0.71 | 0.80 |
| MAIAC TERRA | 1 km | 0.91 | 0.90 | 0.71 | 0.80 |
| MODIS AQUA | 0.05° | 0.77 | 0.50 | 0.30 | 0.37 |
| MODIS TERRA | 0.05° | 0.81 | 0.65 | 0.32 | 0.43 |
| NISE | 25 km | 0.85 | 0.87 | 0.37 | 0.51 |

Table A2: Evaluation of daily snow extent data set performance for 2015. GHCN-D surface
observations are used as "truth". The highest value for each metric is shown in bold.

|  | Accuracy | Precision | Recall | F |
|---|---|---|---|---|
| CMC | 0.92 | 0.81 | 0.81 | 0.81 |
| IMS | **0.93** | 0.84 | **0.85** | **0.84** |
| MAIAC AQUA | 0.87 | 0.69 | 0.73 | 0.71 |
| MAIAC TERRA | 0.88 | 0.68 | 0.73 | 0.71 |
| MODIS AQUA | 0.78 | 0.50 | 0.41 | 0.45 |
| MODIS TERRA | 0.83 | 0.68 | 0.43 | 0.53 |
| NISE | 0.85 | **0.87** | 0.37 | 0.52 |

Table A3: Evaluation of daily snow extent data set performance for 2015. GHCN-D surface
observations are used as "truth". All products are regridded to a common 25 km resolution. The
highest value for each metric is shown in bold.