# Peer review of "Assessing snow extent data sets over North America to inform and improve trace gas"

_Atmospheric Measurement Techniques, 2018_

## Referee Comment (RC1) · Anonymous Referee #1 · 19 Feb 2018

General comments

The paper deal with a very interesting topic, often neglected in trace gas retrievals: the role of snow-covered surface. The paper evaluates several snow database to identify the most appropriate for trace gas retrievals, especially focusing on NO2 retrievals from current and future missions. The authors also point out the potential of the increased sensitivity to NO2 signal over snow-covered surfaces. I have a few suggestions for improvement that I listed below. I recommend the publication after addressing the following comments:

Specific comments

1. In the section describing IMS-dataset you might want to explain a bit more in detail what instruments the dataset is based on.

2. There is a fractional snow extent product from Globsnow/Sen3app projects that might be also worth a look and included in the comparison. For 2015 it is based on VIIRS (Suomi-NPP) data. The data and information are available here: http://www.globsnow.info/index.php?page=SE or here: http://sen3app.fmi.fi/index.php?page=Fractional_Snow_Cover_Extent_-_NH&style=main

3. In the conclusion you write: "However, the lack of confidence in snow identification has previously led many retrieval procedures to omit observations over snow. Increasing this confidence such that these observations could be included would not only improve spatial and temporal sampling, but also allow the inclusion of observations with higher quality information on the lower troposphere." It would be useful to actually demonstrate this with an example or case study, perhaps based on OMI data. I mean, showing one OMI scene/orbit of NO2 retrievals, where the added value of this improved snow information would be visible. For example, an OMI orbit with snow-cover that was filtered out or somehow incorrectly flagged and would be improved using a more accurate knowledge of the snow cover (with the right AMFs and profiles) in the NO2 retrieval.

4. Could you comment on how the increased sensitivity in the PBL might affect NO2 retrievals at relatively higher latitudes (where snow is very often present)? For example, how would those scattering weight profiles in Fig. 2 look like for higher SZA/or a different latitude? It might be less important for TEMPO but it is relevant for OMI/TROPOMI missions to improve retrieval at high latitudes in autumn-winter.

5. There is this paper by Vasilkov et al. about BRDF and OMI retrievals you might need to mention/discuss in your paper: Vasilkov, A., Qin, W., Krotkov, N., Lamsal, L., Spurr, R., Haffner, D., Joiner, J., Yang, E.-S., and Marchenko, S.: Accounting for the effects

of surface BRDF on satellite cloud and trace-gas retrievals: a new approach based on geometry-dependent Lambertian equivalent reflectivity applied to OMI algorithms, Atmos. Meas. Tech., 10, 333-349, https://doi.org/10.5194/amt-10-333-2017, 2017.

---

## Referee Comment (RC2) · Anonymous Referee #2 · 23 Feb 2018

Review of "Assessing snow extent data sets over North America to inform trace gas retrievals from solar backscatter" by M.J. Cooper et al.

In the paper, different snow cover data sets are evaluated to identify which is most suitable for TEMPO trace gas algorithms. Additionally, the authors examine the NO2 AMF sensitivity to surface reflectance using radiative transfer (RT) simulations. The paper contains significant original material that can be of interest for the developers of trace gas algorithms for satellite sensors. The paper subject is appropriate to AMT. The abstract provides a sufficiently complete summary of the paper. The manuscript is well organized. The paper can be recommended for publication after the authors address

the following comments.

General comments

1. The assessment of different snow cover data sets is carried out for the entire year of 2015. This approach of using the full year data may cause biases in the metrics. The authors admit "All data sets have high accuracy numbers, owing to a high number of true negatives during the summer months" (Line 220). I think that the assessment of the snow cover data sets should be done on a seasonal basis and the metrics for different seasons should be compared. It would be particularly interesting to assess the snow data sets for spring when melting snow occurs.

2. In my opinion, results of the RT simulations shown in Fig. 2 and 3 do not provide new significant information. Effects of surface reflectance on trace gas retrievals have been studied theoretically (see O'Bryne et al., JGR, 2010; Lin et al., ACP, 2015; Vasilkov et al., 2017 and references there). Figure 2 of the manuscript (showing the scattering weights for a single solar zenith angle and a single NO2 profile) is not conclusive because the NO2 sensitivity to surface reflectance substantially depends on tropospheric NO2 profiles (see Fig. 13 in Vasilkov et al., AMT, 2017). Figure 3 compares AMFs for snow-covered and snow-free conditions for January 2013. The snow-free conditions are absolutely unrealistic for January. That is why I doubt that useful information can be derived from this comparison. I think that the text and figures related to the RT simulations can be removed without the loss of significant material. To some extent, this is supported by the title of the manuscript because the RT simulations are not mentioned in the title.

Specific comments

Line 24. The quantity "F" is not defined here.

Line 52. It is worthwhile to mention that uncertainties in surface reflectance also lead to uncertainties in the cloud fraction and pressure retrievals which affect the NO2 retrievals (Vasilkov et al., AMT, 2017).

Line 162. Indeed, snow reflectivity is almost spectrally independent in UV/Vis. However, the maps in Fig. 1 include snow-free regions. For such regions, ground reflectivity does depend on wavelength, so reflectivity at 354 nm may not be used for 440 nm.

Line 174. Please clarify "the most reliable source is used".

Line 185. Please explain why the F score is most relevant for TEMPO.

Line 190. Where does the OMI cloud fraction come from? How is the cloud fraction determined for snow-covered and partially snow-covered scenes?

Line 235. Is it correct that the MODIS products perform better at coarser resolution? Table 1 shows F=0.46 and 0.54 for the 4 km resolution while Table 2A shows F=0.45 and 0.53 for the 25 km resolution.

Reference to McLinden et al., ACP, 2014 is missing.

Figure 1. The capture states "reflectivity at visible wavelengths". The 354 nm wavelength (used for the upper panel) is not a visible wavelength. The lower panel is not informative because the color scale is not appropriate for it.

Figure 2. The corresponding NO2 profiles should be shown. Surface reflectivities should be specified. What is the viewing zenith angle of observations?

Appendix. Please explain why some numbers for the CMC and NISE data sets are slightly different in Tables A1 and A2. The special resolution of the data sets is same for both tables.

---

## Author Comment (AC2) · 11 Apr 2018

Thank you for your comments.

——- General Comments

Comment 1: "The assessment of different snow cover data sets is carried out for the entire year of 2015. This approach of using the full year data may cause biases in the metrics. The authors admit "All data sets have high accuracy numbers, owing to a high number of true negatives during the summer months" (Line 220). I think that the assessment of the snow cover data sets should be done on a seasonal basis and the

metrics for different seasons should be compared. It would be particularly interesting to assess the snow data sets for spring when melting snow occurs."

Response: We have included a table in the Appendix that gives evaluations of the snow data sets by season, and now include the following text on Line 245:

"Data sets were also evaluated by season with similar results (Appendix Table A1). All data sets have weaker performance metrics during the spring melt season, which has been observed in past evaluations (Frei et al., 2012). IMS has the highest F score in winter and autumn but is slightly outperformed by MAIAC in spring."

However, keeping in mind that the goal of this work is to evaluate data sets for informing retrieval algorithms, and as most retrieval algorithms would likely choose a single data set to provide snow information throughout the year, we continue to focus on the full year data.

-

Comment 2: "In my opinion, results of the RT simulations shown in Fig. 2 and 3 do not provide new significant information. Effects of surface reflectance on trace gas retrievals have been studied theoretically (see O'Bryne et al., JGR, 2010; Lin et al., ACP, 2015; Vasilkov et al., 2017 and references there). "

Response: We respectfully contend that Figures 2-3 do provide important information here. They illustrate how changes in snow cover affect the observation sensitivity to NO2. Indeed Reviewer 1 expressed interest in Figure 2.

Comment: "Figure 2 of the manuscript (showing the scattering weights for a single solar zenith angle and a single NO2 profile) is not conclusive because the NO2 sensitivity to surface reflectance substantially depends on tropospheric NO2 profiles (see Fig. 13 in Vasilkov et al., AMT, 2017). "

Response: It is true that the column NO2 sensitivity depends on tropospheric NO2 pro-files. However, the scattering weights in Figure 2 represent the sensitivity of backscattered radiation to surface reflectance, which is independent of NO2 profile. We have taken care to clarify this in the text (Line 201):

"Figure 2 shows the sensitivity of backscattered radiation (scattering weights) over snow-covered and snow-free surfaces . . ."

Comment: "Figure 3 compares AMFs for snow-covered and snow-free conditions for January 2013. The snow-free conditions are absolutely unrealistic for January. That is why I doubt that useful information can be derived from this comparison."

Response: We clarified that the figure is for the observation geometry of January. The Figure 3 "snow-free conditions" plot shows AMF values in the case that snow is not present during a given observation. It is not meant to suggest that snow is never present in January in North America. As snow-covered scenes are often omitted in retrieval algorithms, the resulting data sets are essentially "snow-free", and thus a snow-free map of AMF does provide important context.

Comment: "I think that the text and figures related to the RT simulations can be removed without the loss of significant material. To some extent, this is supported by the title of the manuscript because the RT simulations are not mentioned in the title."

Response: We have strengthened the material covering snow and AMFs by including Figure 6, which shows how including snow-covered scenes improves the quantity and quality of retrieval data sets. We have changed the title to reflect this as well. Together with Figures 2-3 we feel that this is new, significant information.

––– Specific comments

Comment: "Line 24. The quantity "F" is not defined here."

Response: We have removed the mention of F here. It is defined later in the abstract.

-

Comment: "Line 52. It is worthwhile to mention that uncertainties in surface reflectance

also lead to uncertainties in the cloud fraction and pressure retrievals which affect the NO2 retrievals (Vasilkov et al., AMT, 2017). "

Response: We now mention this effect in the introduction (Line 59): "Correspondingly, surface snow may be mistaken for cloud, leading to errors in cloud fraction and pressure estimates used in trace gas retrievals (O'Byrne et al., 2010; Lin et al., 2015; Vasilkov et al., 2017)."

-

Comment: "Line 162. Indeed, snow reflectivity is almost spectrally independent in UV/Vis. However, the maps in Fig. 1 include snow-free regions. For such regions, ground reflectivity does depend on wavelength, so reflectivity at 354 nm may not be used for 440 nm.

Response: The snow reflectivity (for 354 nm) is only used when snow is present. Snow-free regions use the MODIS CMG Gap-Filled Snow-Free Products at 470 nm, which are at a wavelength closer to the 440 nm used in the AMF calculation. We have clarified this in the text and in Figure 1.

-

Comment: "Line 174. Please clarify "the most reliable source is used". "

Response: As stated, the GHCN-D data set includes information from multiple sources. GCHN-D provides a priority ranking of these sources. We have added a citation to this line which provides additional information.

-

Comment: "Line 185. Please explain why the F score is most relevant for TEMPO."

Response: This is now clarified in the text (Line 192) as follows: "The F score balances recall (which accounts for false negatives) and precision (which accounts for false positives) to measure correct classification of snow without the influence of frequent snow-free periods, and therefore is the metric which is most relevant for TEMPO"

-

Comment: "Line 190. Where does the OMI cloud fraction come from? How is the cloud fraction determined for snow-covered and partially snow-covered scenes? "

Response: We no longer use the OMI cloud fraction in this work. From line 199: "We assume cloud-free conditions in the AMF calculations, as the impact of surface reflectance on retrieved cloud fractions is beyond the scope of this paper."

-

Comment: "Line 235. Is it correct that the MODIS products perform better at coarser resolution? Table 1 shows F=0.46 and 0.54 for the 4 km resolution while Table 2A shows F=0.45 and 0.53 for the 25 km resolution."

Response: Yes, MODIS products do perform better when regridded to 4km than at their native resolution of $0.05°$, where F=0.37 and 0.43. However as pointed out by the reviewer, the benefit of regridding does not continue to improve if the resolution is further decreased. This has been clarified in the text (Line 250):

"...MODIS Aqua and Terra products perform better when regridded from their native $0.05°$ resolution to a 4 km resolution as it reduces the number of grid boxes missing observations due to cloud..."

-

Comment: "Reference to McLinden et al., ACP, 2014 is missing."

Response: This has been fixed.

-

Comment: "Figure 1. The caption states "reflectivity at visible wavelengths". The 354 nm wavelength (used for the upper panel) is not a visible wavelength. The lower panel

is not informative because the color scale is not appropriate for it. "

Response: The figure caption now specifies "UV-Visible" instead of only "visible" wavelengths. We have also changed the colour scale.

-

Comment: "Figure 2. The corresponding NO2 profiles should be shown. Surface reflectivities should be specified. What is the viewing zenith angle of observations? "

Response: Surface reflectivities and zenith angles are now included in Figure 2. We have edited the text at Line 201 to better distinguish between the sensitivity of backscattered radiation to lower troposphere NO2 (i.e. scattering weights) and the sensitivity of the NO2 column to lower troposphere NO2 (i.e. AMFs). Figure 2 focuses on how the scattering weights themselves (which do not depend on the NO2 profile) are affected by reflectivity, and thus we do not include the corresponding NO2 profiles for the sake of clarity.

"Figure 2 shows the sensitivity of backscattered radiation (scattering weights) over snow-covered and snow-free surfaces for two locations ... This shows that satellite observed backscattered radiation is up to five times as sensitive to NO2 in the boundary layer in the presence of snow, due to the increased absorption by NO2 in the lower troposphere when the surface reflects more sunlight."

-

Comment: "Appendix. Please explain why some numbers for the CMC and NISE data sets are slightly different in Tables A1 and A2. The spatial resolution of the data sets is same for both tables."

Response: Thank you for noticing this. There were some errors in the Appendix tables that have been corrected. In Table A3 (previously A2), all products were regridded to a common 25km resolution. For NISE, this is slightly different than its native 25km grid, hence a small difference in its F score (0.51 to 0.52).

---

## Author Response (AR1)

**RESPONSE TO REVIEWER 1**

Thank you for your comments.

**1. In the section describing IMS-dataset you might want to explain a bit more in detail what instruments the dataset is based on.**

IMS uses an often-changing list of instruments and models to build its dataset. We have added some examples of instruments that are used in Section 2.1.1.

Line 100: "The maps are produced by a trained analyst using visible imagery from a collection of geostationary (e.g. GOES, MeteoSat) and polar orbiting (e.g. AVHRR, MODIS, SAR) satellite instruments, with additional information from microwave sensors (e.g. DMSP, AMSR, AMSU), surface observations (e.g. SNOTEL), and models (e.g. SNODAS) (Helfrich et al., 2007)."

**2. There is a fractional snow extent product from Globsnow/Sen3app projects that might be also worth a look and included in the comparison. For 2015 it is based on VIIRS (Suomi-NPP) data. The data and information are available here: http://www.globsnow.info/index.php?page=SE or here: http://sen3app.fmi.fi/index.php?page=Fractional_Snow_Cover_Extent_-_NH&style=main**

We have looked at the fractional snow extent product from Globsnow/Sen3app as suggested, and have decided to exclude it from this work. This product does not provide snow cover information when clouds are present in the VIIRS observations. As a result, there is no information on snow cover for approximately a third of the TEMPO domain in 2015. Therefore, the product is not appropriate for the study performed here.

**3. In the conclusion you write: "However, the lack of confidence in snow identification has previously led many retrieval procedures to omit observations over snow. Increasing this confidence such that these observations could be included would not only improve spatial and temporal sampling, but also allow the inclusion of observations with higher quality information on the lower troposphere." It would be useful to actually demonstrate this with an example or case study, perhaps based on OMI data. I mean, showing one OMI scene/orbit of NO2 retrievals, where the added value of this improved snow information would be visible. For example, an OMI orbit with snow-cover that was filtered out or somehow incorrectly flagged and would be improved using a more accurate knowledge of the snow cover (with the right AMFs and profiles) in the NO2 retrieval.**

Thank you for this suggestion. We have included a figure (Figure 6) that shows how including observations over snow improves sampling and increases AMFs. This is explained in the text on Line 280 as follows:

"We next examine the effect on both spatial sampling and sensitivity to the lower troposphere of a retrieval data set if observations with surface snow are included rather than omitted. We use IMS to identify the presence of snow for OMI observations over North America in January 2015. We then use LIDORT to calculate AMFs for these observations using the corresponding snow-free (Sun et al., 2017) or snow-covered (O'Byrne et al., 2010) surface reflectance, and examine the results of either including or omitting snow-covered scenes. Figure 6 shows that including snow-covered scenes results in a significant (factor of 2.1) increase in observation frequency, particularly in the northern US and Canada. Additionally, including snow-covered scenes increases the average AMF by a factor of 2.7 in regions with occasional snow cover. The increase in AMF demonstrates that including snow-covered scenes increases the quality of information about the tropospheric $NO_2$ column by increasing the observation sensitivity to tropospheric $NO_2$."

**4. Could you comment on how the increased sensitivity in the PBL might affect NO2 retrievals at relatively higher latitudes (where snow is very often present)? For example, how would those scattering weight profiles in Fig. 2 look like for higher SZA/or a different latitude? It might be less important for TEMPO but it is relevant for OMI/TROPOMI missions to improve retrieval at high latitudes in autumn-winter.**

We have added a scattering weight profile for a high latitude location in Figure 2.

**5. There is this paper by Vasilkov et al. about BRDF and OMI retrievals you might need to mention/discuss in your paper: Vasilkov, A., Qin, W., Krotkov, N., Lamsal, L., Spurr, R., Haffner, D., Joiner, J., Yang, E.-S., and Marchenko, S.: Accounting for the effects of surface BRDF on satellite cloud and trace-gas retrievals: a new approach based on geometry-dependent Lambertian equivalent reflectivity applied to OMI algorithms, Atmos. Meas. Tech., 10, 333-349, https://doi.org/10.5194/amt-10-333-2017, 2017.**

We have added a mention to this paper in the introduction (Line 59):

"Correspondingly, surface snow may be mistaken for cloud, leading to errors in cloud fraction and pressure estimates used in trace gas retrievals (Lin et al., 2015; O'Byrne et al., 2010; Vasilkov et al., 2017)."

and in the conclusion, as follows (Line 316):

"This could potentially include Bidirectional Reflectance Distribution Functions (BRDF) that describe reflection at different viewing angles, as this effect has been shown to have significant impact on retrieved $NO_2$ columns (Vasilkov et al., 2017)"

**RESPONSE TO REVEIWER 2**

Thank you for your comments.

**1. The assessment of different snow cover data sets is carried out for the entire year of 2015. This approach of using the full year data may cause biases in the metrics. The authors admit "All data sets have high accuracy numbers, owing to a high number of true negatives during the summer months" (Line 220). I think that the assessment of the snow cover data sets should be done on a seasonal basis and the metrics for different seasons should be compared. It would be particularly interesting to assess the snow data sets for spring when melting snow occurs.**

We have included a table in the Appendix that gives evaluations of the snow data sets by season, and now include the following text on Line 245:

"Data sets were also evaluated by season with similar results (Appendix Table A1). All data sets have weaker performance metrics during the spring melt season, which has been observed in past evaluations (Frei et al., 2012). IMS has the highest *F* score in winter and autumn but is slightly outperformed by MAIAC in spring."

However, keeping in mind that the goal of this work is to evaluate data sets for informing retrieval algorithms, and as most retrieval algorithms would likely choose a single data set to provide snow information throughout the year, we continue to focus on the full year data.

**2. In my opinion, results of the RT simulations shown in Fig. 2 and 3 do not provide new significant information. Effects of surface reflectance on trace gas retrievals have been studied theoretically (see O'Bryne et al., JGR, 2010; Lin et al., ACP, 2015; Vasilkov et al., 2017 and references there).**

We respectfully contend that Figures 2-3 do provide important information here. They illustrate how changes in snow cover affect the observation sensitivity to $NO_2$. Indeed Reviewer 1 expressed interest in Figure 2.

**Figure 2 of the manuscript (showing the scattering weights for a single solar zenith angle and a single NO2 profile) is not conclusive because the NO2 sensitivity to surface reflectance substantially depends on tropospheric NO2 profiles (see Fig. 13 in Vasilkov et al., AMT, 2017).**

It is true that the *column $NO_2$ sensitivity* depends on tropospheric $NO_2$ profiles. However, the scattering weights in Figure 2 represent the *sensitivity of backscattered radiation* to surface reflectance, which is independent of $NO_2$ profile. We have taken care to clarify this in the text (Line 201):

"Figure 2 shows the sensitivity of backscattered radiation (scattering weights) over snow-covered and snow-free surfaces …"

**Figure 3 compares AMFs for snow-covered and snow-free conditions for January 2013. The snow-free conditions are absolutely unrealistic for January. That is why I doubt that useful information can be derived from this comparison.**

We clarified that the figure is for the observation geometry of January. The Figure 3 "snow-free conditions" plot shows AMF values in the case that snow is not present during a given observation. It is not meant to suggest that snow is never present in January in North America. As snow-covered scenes are often omitted in retrieval algorithms, the resulting data sets are essentially "snow-free", and thus a snow-free map of AMF does provide important context.

**I think that the text and figures related to the RT simulations can be removed without the loss of significant material. To some extent, this is supported by the title of the manuscript because the RT simulations are not mentioned in the title.**

We have strengthened the material covering snow and AMFs by including Figure 6, which shows how including snow-covered scenes improves the quantity and quality of retrieval data sets. We have changed the title to reflect this as well. Together with Figures 2-3 we feel that this is new, significant information.

**Specific comments Line 24. The quantity "F" is not defined here.**

We have removed the mention of $F$ here. It is defined later in the abstract.

**Line 52. It is worthwhile to mention that uncertainties in surface reflectance also lead to uncertainties in the cloud fraction and pressure retrievals which affect the NO2 retrievals (Vasilkov et al., AMT, 2017).**

We now mention this effect in the introduction (Line 59):

"Correspondingly, surface snow may be mistaken for cloud, leading to errors in cloud fraction and pressure estimates used in trace gas retrievals (O'Byrne et al., 2010; Lin et al., 2015; Vasilkov et al., 2017)."

**Line 162. Indeed, snow reflectivity is almost spectrally independent in UV/Vis. However, the maps in Fig. 1 include snow-free regions. For such regions, ground reflectivity does depend on wavelength, so reflectivity at 354 nm may not be used for 440 nm.**

The snow reflectivity (for 354 nm) is only used when snow is present. Snow-free regions use the MODIS CMG Gap-Filled Snow-Free Products at 470 nm, which are at a wavelength closer to the 440 nm used in the AMF calculation. We have clarified this in the text and in Figure 1.

**Line 174. Please clarify "the most reliable source is used".**

As stated, the GHCN-D data set includes information from multiple sources. GCHN-D provides a priority ranking of these sources. We have added a citation to this line which provides additional information.

**Line 185. Please explain why the F score is most relevant for TEMPO.**

This is now clarified in the text (Line 192) as follows:

"The $F$ score balances recall (which accounts for false negatives) and precision (which accounts for false positives) to measure correct classification of snow without the influence of frequent snow-free periods, and therefore is the metric which is most relevant for TEMPO"

**Line 190. Where does the OMI cloud fraction come from? How is the cloud fraction determined for snow-covered and partially snow-covered scenes?**

We no longer use the OMI cloud fraction in this work. From line 199:

"We assume cloud-free conditions in the AMF calculations, as the impact of surface reflectance on retrieved cloud fractions is beyond the scope of this paper."

**Line 235. Is it correct that the MODIS products perform better at coarser resolution? Table 1 shows F=0.46 and 0.54 for the 4 km resolution while Table 2A shows F=0.45 and 0.53 for the 25 km resolution.**

Yes, MODIS products do perform better when regridded to 4km than at their native resolution of $0.05°$, where F=0.37 and 0.43. However as pointed out by the reviewer, the benefit of regridding does not continue to improve if the resolution is further decreased. This has been clarified in the text (Line 250):

"…MODIS Aqua and Terra products perform better when regridded from their native $0.05°$ resolution to a 4 km resolution as it reduces the number of grid boxes missing observations due to cloud…"

**Reference to McLinden et al., ACP, 2014 is missing.**

This has been fixed.

**Figure 1. The caption states "reflectivity at visible wavelengths". The 354 nm wavelength (used for the upper panel) is not a visible wavelength. The lower panel is not informative because the color scale is not appropriate for it.**

The figure caption now specifies "UV-Visible" instead of only "visible" wavelengths. We have also changed the colour scale.

**Figure 2. The corresponding NO2 profiles should be shown. Surface reflectivities should be specified. What is the viewing zenith angle of observations?**

Surface reflectivities and zenith angles are now included in Figure 2. We have edited the text at Line 201 to better distinguish between the sensitivity of backscattered radiation to lower troposphere $NO_2$ (i.e. scattering weights) and the sensitivity of the $NO_2$ column to lower troposphere $NO_2$ (i.e. AMFs). Figure 2 focuses on how the scattering weights themselves (which do not depend on the $NO_2$ profile) are affected by reflectivity, and thus we do not include the corresponding $NO_2$ profiles for the sake of clarity.

"Figure 2 shows the sensitivity of backscattered radiation (scattering weights) over snow-covered and snow-free surfaces for two locations … This shows that satellite observed backscattered radiation is up to five times as sensitive to $NO_2$ in the boundary layer in the presence of snow, due to the increased absorption by $NO_2$ in the lower troposphere when the surface reflects more sunlight."

**Appendix. Please explain why some numbers for the CMC and NISE data sets are slightly different in Tables A1 and A2. The spatial resolution of the data sets is same for both tables.**

Thank you for noticing this. There were some errors in the Appendix tables that have been corrected. In Table A3 (previously A2), all products were regridded to a common 25km resolution. For NISE, this is slightly different than its native 25km grid, hence a small difference in its $F$ score (0.51 to 0.52).

[revised manuscript text omitted]

---

## Author Response (AR2)

**1. Your statements that trace gas retrievals will improve when including snow-covered scenes are too strong. I agree that the high surface reflectivity is potentially benefitting the vertical sensitivity to NO2 in clear-sky situations. But the reality for all retrievals is that they need highly accurate cloud retrieval schemes that can distinguish between a bright surface and bright, low altitude clouds. I don't think we have those in place yet. I think this line of thought should be reflected in the text.**

Thank you for this suggestion. We have added a paragraph in the conclusion to discuss cloud retrievals as follows (Line 325):

"The potential improvements in $NO_2$ retrieval performance over snow-covered scenes outlined here were tested for clear-sky conditions. The accuracy of cloud retrieval schemes also impacts the quality of trace gas retrievals. Many cloud retrieval schemes have difficulty distinguishing between a bright surface and bright, low altitude clouds; This may diminish the impact that improved surface snow reflectance can have on observation frequency and sensitivity when clouds are present. However, using accurate surface snow cover information may also lead to corresponding improvements in cloud retrieval accuracy."

**P7, L14: …high reflectance is potentially advantageous …**

This has been changed.

**P8, L36-39: clarify that the number of observations and sensitivity improves strictly in clear-sky situations, in reality snow-covered scenes will often be covered by clouds screening the NO2 below.**

These lines now read:

"We find that using IMS to identify snow cover and enable inclusion of snow-covered scenes in clear-sky conditions across North America in January..."

**P17, L295-297: "The increase in AMF … to tropospheric NO2." is a too strong statement. You should clarify that this is only valid if one assumes that a cloud algorithm will handle the snow-covered scenes with 100% reliability. The quality of information will furthermore only be achieved if the NO2 profile and albedo value are of high quality.**

We now moderate this statement as follows:

"As we assume clear-sky conditions, these are likely upper bounds on potential increases in observation quantity and quality. In practice, the presence of clouds and errors in cloud retrieval algorithms will likely diminish these impacts."

**2. You test in the paper the presence of snow – but what about the snow albedo value? In Figure 4, these values are listed as 0.81 and 0.83, but based on what? Adding a bit of**

**discussion on the snow albedo values from the various sets and their uncertainties would be useful.**

As stated in Section 2.3, snow reflectivity values used here come from the product described in O'Byrne et al. (2010). We have added information on uncertainty as follows (Line 171):

"This data set is consistent with previous snow reflectivity (e.g. Moody et al., 2007; Tanskanen and Manninen, 2007) over most land types (O'Byrne et al., 2010). Snow-covered reflectivity has an estimated uncertainty of 10-20% in most regions, with higher uncertainties in regions with thin or transient snow."

**P14, L211: please rephrase considering that snow albedo values may be quite uncertain and that residual clouds may be present affecting this sensitivity – otherwise the readers may wrongfully conclude "snow means better sensitivity", whereas the message should be that "under clear-sky conditions and with very good knowledge of snow albedo, the presence of snow means better sensitivity".**

This has been rephrased as follows:

"This shows that satellite observed backscattered radiation in clear-sky conditions is up to five times as sensitive to $NO_2$ in the boundary layer after accounting for increased reflection by snow, due to the increased absorption by $NO_2$ in the lower troposphere when the surface reflects more sunlight"

We also added a sentence in the conclusion that mentions the importance of good knowledge of snow albedo (Line 336):

"Accurate knowledge of snow reflectivity is also needed to improve retrievals over snow."

**Specific comments:**

**P9, L65: please add that … are often omitted or flagged as unreliable to avoid potential errors**

This has been changed.

**P17, L302: please add …retrievals over clear-sky reflective snow-free surfaces…**

This has been changed.

**P18, L326: please add …significant impact on retrieved NO2 columns and clouds (Vassilkov et al., 2017; Lorente et al., 2018).**

This has been changed.

**Caption Figure 3: please explain how you obtained the snow reflectance value.**

This has been changed. Caption now states:

[revised manuscript text omitted]